# The Mental Health and Syndemic Effect on Suicidal Ideation among Migrant Workers in China: A Cross-Sectional Study

**DOI:** 10.3390/ijerph182111363

**Published:** 2021-10-29

**Authors:** Kechun Zhang, Chen Xu, Yinghuan Zhang, Rongxi Wang, Xiaoyue Yu, Tian Hu, Yaqi Chen, Zixin Wang, Bolin Cao, Hui Chen, Yujie Liu, Shangbin Liu, Huachun Zou, Yong Cai

**Affiliations:** 1Shenzhen Longhua District Center for Disease Control and Prevention, Shenzhen 518000, China; zkc1317@yeah.net (K.Z.); ht1137571641@126.com (T.H.); chloe4697@163.com (Y.C.); 2School of Public Health, Shanghai Jiao Tong University School of Medicine, Shanghai 201800, China; xuchen233333@163.com (C.X.); zhangyinghuan0516@163.com (Y.Z.); RosieW816@outlook.com (R.W.); dd2192003@163.com (X.Y.); chenhui711@yeah.net (H.C.); liuyj_4287@163.com (Y.L.); liushangbin12139@163.com (S.L.); 3JC School of Public Health and Primary Care, Faculty of Medicine, The Chinese University of Hong Kong, Hong Kong 999077, China; wangzx@cuhk.edu.hk; 4School of Media and Communication, Shenzhen University, Shenzhen 518000, China; caobolin@szu.edu.cn; 5School of Public Health (Shenzhen), SunYat-sen University, Shenzhen 518107, China; 6Kirby Institute, University of New South Wales, Sydney 2052, Australia

**Keywords:** migrant workers, suicidal ideation, mental health, syndemic, China

## Abstract

Background: Migrant workers are in a socially disadvantaged position and thus suffer from more stress and mental health disorders, resulting in a high risk of suicidal ideation. This study aimed to explore the association between psychosocial problems and suicidal ideation, and the syndemic effect of concurrent psychosocial problems on suicidal ideation among migrant workers. Methods: We conducted a cross-sectional study and recruited 1805 migrant workers in Shenzhen, China. Each participant completed a self-administered questionnaire to report sociodemographic information and mental health status. Univariate and multivariate logistic regression were used to explore the association between psychosocial variables and suicidal ideation, and their syndemic effect on suicidal ideation. Results: The prevalence of suicidal ideation among migrant workers was 7.5%. All selected psychosocial variables were independently associated with suicidal ideation. Multivariate logistic regression analysis showed that three psychosocial variables were associated with suicidal ideation: anxiety (ORm: 1783, 95% CI: 1.089–2.920), entrapment (ORm: 2.064, 95% CI: 1.257–3.388), and defeat (ORm: 2.572, 95% CI: 1.612–4.103). Various mental health issues can exist simultaneously to increase the risk of suicidal ideation (AOR: 5.762, 95% CI: 3.773–8.802). Workers with more psychosocial problems were more likely to have suicidal ideation. Conclusions: The association between poor mental health and suicidal ideation should not be overlooked among migrant workers. The co-occurring or syndemic effect of psychosocial problems may increase the risk of suicidal ideation.

## 1. Introduction

A migrant is defined as any person moving across an international border or within a country away from his/her habitual place of residence for work or study, for escaping conflict or persecution, or for environmental factors [1,2]. Migration is definitely a worldwide phenomenon, as the global population of international migration had reached 281 million in 2020 [3]. Nearly two thirds of those were labor migrants [1]. In China, there were about 28.56 million migrant workers in 2020 [4], accounting for a considerable proportion of the total population. Compared with local residents, migrant workers are at a disadvantage in some aspects such as health insurance, labor security, social welfare, housing subsidies, and children’s education due to the administrative barriers of China’s household registration system (hukou) which plays an important role in social resource allocation [5,6]. Therefore, migrant workers have long been a vulnerable group of Chinese society. There was a suicide cluster event at Foxconn’s Shenzhen factory between January and May 2010. The 12 victims were all migrant workers, which had sparked national concern about the suicide status of migrant workers [7].

As we all know, suicide is a serious public health problem and a leading cause of death and disability worldwide [8]. In the suicide theory, the development of suicidal ideation and the progression from ideation to suicide attempts are both predictors of suicide deaths [8]. Suicidal ideation, a prodrome of suicide completion [9,10] is defined as thinking about or considering suicide [8]. An individual who had expressed suicidal ideation has a higher risk of subsequent completed suicide than people who had not [11]. The lifetime prevalence of suicidal ideation among Chinese migrant workers was up to 12.8% [12], which was about three times higher than the prevalence in the general population (3.9%) [9]. Chen et al. considered migration status as a reason for the high prevalence [12]. In order to reduce suicide risk among Chinese migrant workers, early identification of individuals experiencing suicidal ideation is important.

It is becoming clear that most mental disorders such as depression, hopelessness, anxiety, and impulsivity are conceptualized as predictors of suicidal ideation [8,13,14,15,16]. Research has found that depression and anxiety were the most important risk factors of suicidal ideation [17]. A systematic review revealed that exposure to different job stressors brought about elevated odds of suicidal ideation to varying degrees [18]. Entrapment, defined as a feeling where an individual has a strong motivation to escape from an unbearable situation but is incapable of escape [19], may be a predictor of suicidal ideation [20]. Measuring individuals’ perceptions of failed struggle and losing rank, defeat can also be a predictor of suicidal ideation [21]. According to the interpersonal theory of suicide, perceived burdensomeness and thwarted belongingness (i.e., interpersonal needs) were linked with suicidal ideation [22]. Based on the previous studies, the roles of mental health problems above in the development of suicidal ideation are worth exploring in migrant workers.

Different mental health problems may co-occur and have a syndemic effect, leading to suicidal ideation. The concept ‘syndemic’ introduced by medical anthropologists in the late twentieth century, is defined as the synergistic interaction of two or more coexistent diseases and the resultant extra disease burden for a population [23]. For the sake of analyzing the biosocial connections in health-related fields, syndemic theory constructs a framework of inter-disease and social condition-disease interactions [24]. A Chinese cross-sectional study among patients with sexually transmitted infections demonstrated that the prevalence of suicidal ideation of patients with two or more psychosocial problems quadrupled that of the non-syndemic group, and it is more severe in groups with five or more psychosocial problems [25]. Similar conclusions were drawn from a study of psychosocial syndemic with suicidal ideation among men who have sex with men [26]. Nevertheless, in terms of migrant workers’ suicidal ideation, there has been no syndemic-relevant research at present.

The purposes of our study were to learn more about mental health status and explore the association between mental health problems as well as their syndemic conditions and suicidal ideation in Chinese migrant workers. We aimed to better understand the psychosocial problems and needs of migrant workers, and to provide a theoretical basis for targeted intervention from a psychosocial perspective and effective reduction of their suicide risk based on following hypothesis:

**Hypothesis 1** **(H1).**
*All selected psychosocial variables including depression, anxiety, occupational stress, entrapment, defeat and interpersonal needs were associated with migrant workers’ suicidal ideation.*


**Hypothesis 2** **(H2).**
*Different psychosocial problems can co-occur and have a syndemic effect on suicidal ideation among migrant workers.*


## 2. Materials and Methods

### 2.1. Participants and Sample Size

A cross-sectional study was carried out from October to December 2019 in Shenzhen, China. Shenzhen, one of the oldest and most successful special economic development zones, has become China’s largest migrant city [27]. A permanent non-registered population has reached 8.491 million in 2019 [28]. Participants met the following inclusion criteria: (1) full-time industrial workers; (2) over the age of 18; (3) living in Shenzhen but from other cities. The sample size was calculated by using the formula to estimate a proportion of the population with a specified relative precision:$$n=Z_{1-\alpha }^{2}\frac{p\left(1-p\right)}{\delta^{2}}$$

The parameter for this calculation included: *α* = 0.05, *p* = 33.6% (expected prevalence of suicidal ideation among migrant workers according to a prior study [29]), *δ* = 0.025 (error tolerance). The required sample size was calculated to be 1371. Considering a non-response rate of 20%, a total of 1713 participants was needed for this study.

### 2.2. Study Procedure

A group of one epidemiologist, one health psychologist, one health communication expert, two public health researchers, and one plant worker developed the survey questionnaire. A pilot study was conducted among 20 factory workers to evaluate the readability and clarity of the questionnaire. According to their comments, the group proceeded with the corresponding revision to form the final questionnaire. The 20 persons participating in pilot study were not included in the follow-up site survey.

A multistage stratified sampling method was used to recruit a representative sample of migrant workers. Considering the average number of workers per workshop was approximately 40 to 50, the research team selected 36 workshops from 16 factories for this study. First, 16 factories (one food and beverage manufacturer, one garment factory, one smelter, two chemical raw materials plants, three printing and dyeing factories, three electronic devices manufacturers, four mechanical processing plants, and one other factory) were randomly selected. Two to three workshops were then randomly selected from each factory. All workers in the selected workshops were subsequently invited to participate in the survey in Longhua District Center for Disease Control and Prevention (CDC).

Prior to the survey, trained fieldworkers firstly ascertained the eligibility of the participants, introduced the purpose and content of the study, and answered their questions and concerns. After written informed consent was obtained, each participant was asked to complete an anonymous self-administered questionnaire in a separate room for protecting their privacy. After completing the questionnaire, each respondent received a cash coupon of ¥20 (US$2.60) for their time.

### 2.3. Ethical Approval

Ethics approval was obtained from the Ethics Committee of School of Public Health, Sun Yat-sen University (March 2019). The written informed consent was obtained from participants.

### 2.4. Measure

#### 2.4.1. Sociodemographic Variables

Sociodemographic information about migrant workers included age group, gender, education level, monthly income, marital status, number of children, work time outside hometown, and occupation.

#### 2.4.2. Psychosocial Variables

Depression: The Centre for Epidemiological Studies Depression Scale (CES-D-10) was a short scale that measures depressive symptoms in the past week in both general and clinical populations [30]. It was comprised of 10 items and three factors including positive affect, depressed affect, and somatic symptoms. The response options for each item ranged from 0 (less than a day or never) to 3 (5–7 days) [31]. Items 5 and 8 were positive affect statements and reversely scored. A higher total score indicated greater severity of symptoms (Cronbach’s α = 0.683; range 0–30). The Chinese version of the scale had been already applied in old people for measuring depressive symptoms [32]. The cutoff score for depressive symptoms was 10 [30].

Anxiety: The Generalized Anxiety Disorder (GAD) Scale, a 7-item scale, was used to measure how often the individuals were bothered by each anxiety symptom during the past two weeks [33]. The response options were “not at all”, “several days”, “more than half the days” and “nearly every day”, scored from 0 to 3, respectively. A higher total score (range 0–21) indicated more severe anxiety symptoms. A reasonable cut point for identifying probable cases of GAD was set to a value of 10. The GAD-7 had been validated in China as a screening instrument to detect the severity of anxiety symptoms [34,35]. The Cronbach’s α coefficient of the scale in this study was 0.917.

Occupational stress: Cooper, C.L designed the Occupational Stress Questionnaire to measure occupational stress. The questionnaire was a well-established measurement tool and applied to occupational mental health researches. It was a 13-item self-reported questionnaire containing four factors, namely, workload (five items), role conflict (three items), role ambiguity (two items), and utilization of skills (three items). Each item was rated from 1 (never) to 7 (always) [36]. The Chinese version of the questionnaire had good measurement reliability and construct validity [37]. The internal consistency of the scale had good reliability, with Cronbach’s α value of 0.718.

Entrapment: The feeling of entrapment from the external and internal world was assessed by the Entrapment Scale (ES) which consisted of 16 items [38]. The response options of each item were assigned a score of 0–4 representing “not at all”, “a little bit”, “moderately”, “quite a bit”, and “extremely”, respectively [39]. The higher the total score, the greater the degree of entrapment (Cronbach’s α = 0.956; range 0–30). The Chinese version of ES proved to be a valid and reliable screening tool in a prior study [40].

Defeat: The level of defeat was evaluated by the Defeat Scale (DS), which was a 16-item and 5-point rating scale, ranging from 0 (never) to 4 (always). The overall score ranged from 0 to 64 (Cronbach’s α = 0.880). The level of the total score corresponded to the severity of the defeat symptoms [38]. The validity and reliability of the DS had been confirmed in different populations in China [41,42].

Interpersonal Needs: The 15-item Interpersonal Needs Questionnaire (INQ-15) was chosen for testing individual interpersonal needs including perceived burdensomeness and thwarted belongingness in the past week [43]. It was a 7-point Likert scale, ranging from 1 (not true for me at all) to 7 (very true for me). A higher score reflected greater perceived burdensomeness and thwarted belongingness (Cronbach’s α = 0.749, range 15–105). The scale had good psychometric properties and had been applied to Chinese migrant workers [44].

Suicidal ideation: The question on suicidal ideation was as follows: “Have you ever thought about committing suicide? (0 = no, 1 = yes)” [9,45].

### 2.5. Statistical Analysis

Sociodemographic information and psychosocial variables about migrant workers were first grouped and then described by frequency and percentage. There were two ways to determine the cutoff values for the high and low levels grouping of psychosocial variables, either by referring to previous literature such as depression and anxiety, or by using the 75th percentile of the total scale score such as occupational stress, entrapment, defeat, interpersonal needs. Univariate and multivariate logistic regressions were conducted to explore the association between sociodemographic and psychosocial factors and suicidal ideation. People with syndemic conditions must have two or more psychosocial problems, which were further divided into two groups of high and low levels based on the concurrent number of psychosocial problems. Individuals with two to three psychosocial problems were classified as the low-level group and those with four or more psychosocial problems were grouped as the high-level group. Then, univariate logistic regression was carried out to examine the association between psychosocial syndemic conditions and suicidal ideation. All statistical analyses were performed using IBM SPSS Statistics for Windows, Version 25.0 (IBM Corp., Armonk, NY, USA).

## 3. Results

### 3.1. Sociodemographic Characteristics

A total of 2023 questionnaires was collected from the 2700 workers in the selected factory workshops. Excluding 218 Shenzhen locals, the final 1805 questionnaires were included in the statistical analysis. The mean age was 32.02, with a standard deviation of 7.908. Sociodemographic characteristics and their association with suicidal ideation were presented in Table 1. Among the participants, 67.3% were male and 58.7% had an education level below high school. More than half of the migrant workers were married (53.6%) and had a child or children (55.2%), and had worked in Shenzhen no more than 10 years (55.5%).

Univariate logistic regression showed that age group, education level, marital status, number of children, and work time outside hometown were significantly associated with suicidal ideation. Suicidal ideation was more likely to develop in the age groups of less than 30 (ORu = 4.694, 95% CI = 2.134–10.322) and 30–40 (ORu = 2.956, 95% CI = 1.330–6.566) than that in the age group of more than 40. Individuals with the education level of college degree or above were more susceptible than those with an education level below high school (ORu = 2.094, 95% CI = 1.136–3.861). Unmarried workers had a higher risk of suicidal ideation than married workers (ORu = 2.919, 95% CI = 1.997–4.268). The workers who had children or worked outside their hometown for more than 20 years were not prone to suicidal ideation.

### 3.2. Psychosocial Variables

The frequency distribution of psychosocial variables among 1805 migrant workers was presented in Table 2. A total of 7.5% (135/1805) of the participants reported ever having suicidal ideation. In addition, 27% were at a high level of depressive symptoms and 9.5% were classified in the group with a high level of anxiety. When using the 75% percentiles as the cutoff values of high and low levels of psychosocial factors, 26.5%, 26.1%, 25.8% and 35.8% felt excessive occupational stress, entrapment, defeat and unsatisfied interpersonal needs, respectively. More than half of those who ever had suicidal ideation reported experiencing depression (56.3%), entrapment (64.4%), defeat (63.0%), and unsatisfied interpersonal needs (54.8%).

### 3.3. The Association between Psychosocial Variables and Suicidal Ideation

As shown in Table 3, all six psychosocial variables were significantly associated with suicidal ideation after adjusting for significant sociodemographic characteristics. High levels of depression (AOR: 3.691, 95% CI: 2.564–5.313), anxiety (AOR: 4.600, 95% CI: 3.005–7.042), occupational stress (AOR: 1.773, 95% CI: 1.222–2.571), entrapment (AOR: 5.396, 95% CI: 3.696–7.876), defeat (AOR: 5.510, 95% CI: 3.788–8.014), and interpersonal needs (AOR: 2.332, 95% CI: 1.622–3.352) were associated with a high risk of suicidal ideation. However, only three psychosocial factors showed significant relation to suicidal ideation in the multivariate logistic regression: anxiety (ORm: 1783, 95% CI: 1.089–2.920), entrapment (ORm: 2.064, 95% CI: 1.257–3.388), and defeat (ORm: 2.572, 95% CI: 1.612–4.103).

### 3.4. The Association between the Number of Syndemic Conditions and Suicidal Ideation

The association between the number of syndemic conditions and suicidal ideation was presented in Table 4. The percentage of migrant workers having two or more psychosocial problems was 38.8% (701/1805), with 25% (451/1805) at a low level (have two to three psychosocial problems) and 13.8% (250/1805) at a high level (have four or more psychosocial problems). Those with syndemic conditions had a nearly sixfold increased risk of suicidal ideation (AOR: 5.762, 95% CI: 3.773–8.802). The low-level group and high-level group indicated a prominent syndemic effect compared with those in the non-syndemic group (have no more than one psychosocial problem), with AOR values of 3.149 (95% CI: 1.912–5.163) and 11.721 (95% CI: 7.322–18.761), respectively.

## 4. Discussion

This study enriched the current literature about the mental health status and the impact of their syndemic circumstances on suicidal ideation among Chinese migrant workers. The results demonstrated that each of the six selected psychosocial variables was independently associated with suicidal ideation, especially the feelings of entrapment and defeat. The most important finding was that coexistent mental health problems could definitely aggravate the suicidal ideation burden from a syndemic perspective. Migrant workers with more psychosocial problems were more likely to have suicidal ideation.

We found that 7.5% of migrant workers had lifetime suicidal ideation. The prevalence was higher than that of the Chinese general population (*p* = 3.9%, 95% CI: 2.5–6.0%) [9]. Meanwhile, this prevalence was higher than those of other developed countries’ workers like the United States (*p* = 3.5%) [46] and South Korea (*p* = 2.8%) [47]. However, the prevalence was much lower than that of Vietnamese industrial workers (*p* = 33.6%) [29]. There were three possible reasons to explain this phenomenon. First, the traditional Chinese culture regarded suicidal behavior as stigmatized and shameful. Influenced by Chinese culture, people may be hesitant to report their suicidal ideation [45]. Second, using only one question to assess suicidal ideation was likely to underestimate its prevalence [25]. Third, the Chinese people’s living standard has been greatly improved in the past four decades for rapid socio-economic development. Consequently, the prevalence of suicidal ideation may be decreased [9].

According to the results, the migrant workers’ age, education level, marital status, family situation, and years of working outside the home may influence the development of suicidal ideation. Young migrant workers may be under more pressure arising from the instability of their careers, the difficulty of adapting to new environments, a lack of social networks, and a lack of parental and extended family support [6]. The individuals who had been married or become fathers/mothers were less likely to have suicidal ideation, indicating the importance of family support. The highly educated workers may have suicidal ideation due to the huge contrast between reality and ideals, which led to broken ideals and despair [7]. Workers who had worked outside their hometown for a long time may be more experienced in dealing with migration stress and enhancing social competence, including social skills, language proficiency, and personal characteristics [48].

Migrant workers are a special group at the bottom of the social ladder in China. Most of them migrate from rural areas to economically developed regions in pursuit of better employment opportunities and living conditions. The new living and working environment require them to adapt quickly to the local culture, dialect and food. They are engaged in repetitive assembly-line work and work overtime to increase their income. The absence of a local hukou prevents them from receiving the same social and medical benefits as local residents. All of the above difficulties place migrant workers on the margins of urban life, resulting in poor mental health outcomes. The tragedy that happened in Foxconn was an alarm that more attention should be paid to the poor mental health of migrant workers [49,50]. Approximately 38.8% reported a combination of mental health problems, which was similar to the prevalence of common mental health problems of Shenzhen’s migrant workers (*p* = 34.4%) [6]. More seriously, the syndemic effect of psychosocial health problems had a negative health outcome of exacerbating the risk of suicidal ideation among migrant workers. Comprehensive assessment of individuals’ multidimensional mental health is beneficial for early identification, follow-up and intervention of suicidal ideation.

It was noteworthy that the feelings of entrapment and defeat played negative roles in developing suicidal ideation and remained significant in the multivariate logistic regression analysis. The concepts of entrapment and defeat were first proposed according to the social rank theory and proven to be promising variables for the study of depression [38]. Entrapment and defeat can be felt following long-term and stressful life events [40]. Evolutionary models suggested the two feels can co-occur to result in the development of mental disorders. Changes in anxiety and depression between time points could be predicted from baseline levels of entrapment and defeat, and vice versa [39]. Furthermore, studies have shown clear and robust correlations between the two constructs and suicidal ideation [51,52,53]. Apart from that, depression and anxiety were identified as the important negative factors in the generation of suicidal ideation in the population as well as Shenzhen’s migrant service workers [17]. Using structural equation model, Zhou et al. [17] reported that anxiety played an indirect role in suicidal ideation through depression. Occupational stress, the most common occupational problem faced by workers [54], was considered as a determinant of common mental disorders and suicidal ideation. The OR ranged from 1.45 (95% CI: 1.01–2.08) for colleague support and poor supervisor to 1.91 (95% CI: 1.22–2.99) for job insecurity [18], which was in line with the OR [1.789 (95% CI: 1.237–2.587)] of occupational stress on the risk of suicidal ideation in this study. Once the desire and capability to commit suicide are present in people, near-lethal and lethal suicide attempts are more likely to occur [55]. Strong empirical evidence supported that the development of suicidal desire can be predicted through measurement of interpersonal needs including thwarted belongingness and perceived burdensomeness to prevent subsequent suicide attempts [44,56]. As expected, all psychosocial variables in our constructed syndemic model not only increased the risk of suicidal ideation individually, but also can coexist to increase the burden of health consequences.

The study had several limitations when interpreting the results. First, it was a cross-sectional study that inevitably hindered the identification of causal relationships between psychosocial factors and suicidal ideation. Secondary, the sample of migrant workers involved only labor-intensive industries and one region of a southern Chinese city, which limited the generalizability of the findings. Third, sociodemographic and psychosocial factors that may influence suicidal ideation may not have been fully included due to the limitation of the questionnaire length and the need to ensure questionnaire quality. Fourth, there were no clear literature and statistical methods to determine the cut-off values for distinguishing between groups of high and low levels for some psychosocial variables such as entrapment and defeat. The cut-off values of those psychosocial variables were considered to be the 75th percentile of the total scale scores by referring to the methodology of other literature. Fifth, the prevalence of suicidal ideation may be underestimated since the suicidal ideation was evaluated by a single item of the questionnaire. Finally, multivariate regression analysis did not fully address the issue of multiple covariates in psychosocial variables, and thus the results of multiple regression analysis need to be interpreted with some reservations.

## 5. Conclusions

Our study confirmed the high prevalence of suicidal ideation and the association between various mental health problems and suicidal ideation among migrant workers. The study gave us an insight that health-related mental health problems can occur simultaneously and have a syndemic effect, which can increase the risk of suicidal ideation. Various psychosocial health conditions can be integrated to form a comprehensive framework for the preliminary identification of suicidal ideation so that interventions can be implemented to stop the ensuing suicide attempts and suicide completion among migrant workers.

## Figures and Tables

**Table 1 ijerph-18-11363-t001:** Sociodemographic characteristics and their association with suicidal ideation.

Sociodemographic Characteristics	Number (%)	Had Suicidal Ideation	
		Number (%)	ORu (95% CI)
Age group (years)			
<30	704 (39.0%)	73 (54.1%)	4.694 (2.134–10.322) **
30–40	810 (44.9%)	55 (40.7%)	2.956 (1.330–6.566) **
>40	291 (16.1%)	7 (5.2%)	Reference
Gender			
Male	1214 (67.3%)	91 (67.4%)	Reference
Female	591 (32.7%)	44 (32.6%)	0.993 (0.683–1.443)
Education level			
below high school	1059 (58.7%)	67 (49.6%)	Reference
High school	585 (32.4%)	52 (38.5%)	1.444 (0.991–2.106)
College degree or above	113 (6.3%)	14 (10.4%)	2.094 (1.136–3.861) *
Unknown	48 (2.7%)	2 (1.5%)	0.644 (0.153–2.709)
Monthly income (RMB)			
<3000	166 (9.2%)	9 (6.7%)	Reference
3000–4999	1099 (60.9%)	79 (58.5%)	1.351 (0.664–2.747)
>4999	484 (26.8%)	44 (32.6%)	1.744 (0.832–3.656)
Unknown	56 (3.1%)	3 (2.2%)	0.987 (0.258–3.783)
Marital status			
Married	967 (53.6%)	43 (31.9%)	Reference
Unmarried	719 (39.8%)	86 (63.7%)	2.919 (1.997–4.268) **
Divorced/ Widowed	47 (2.6%)	3 (2.2%)	1.465 (0.437–4.908)
Unknown	72 (4.0%)	3 (2.2%)	0.934 (0.283–3.089)
Number of children			
0	808 (44.8%)	90 (66.7%)	Reference
1–2	885 (49.0%)	44 (32.6%)	0.417 (0.287–0.607) **
3–5	112 (6.2%)	1 (0.7%)	0.072 (0.010–0.521) **
Work time outside hometown (years)			
<10	1001 (55.5%)	86 (63.7%)	Reference
10–20	674 (37.3%)	45 (33.3%)	0.761 (0.523–1.107)
>20	130 (7.2%)	4 (3.0%)	0.338 (0.122- 0.936) *
Occupation			
Light industries	1063 (58.9%)	88 (65.2%)	Reference
Heavy industries	392 (21.7%)	25 (18.5%)	0.755 (0.476–1.196)
Other industries	350 (19.4%)	22 (16.3%)	0.743 (0.458–1.205)

ORu: Univariate odds ratios. * *p* < 0.05, ** *p* < 0.01.

**Table 2 ijerph-18-11363-t002:** The frequency distribution of psychosocial variables.

Psychosocial Variables	Number (%)	Had Suicidal Ideation Number (%)
Suicidal Ideation		
Yes	135 (7.5%)	/
No	1670 (92.5%)	/
Depression		
High level (score ≥ 10)	487 (27.0%)	76 (56.3%)
Low level (score < 10)	1318 (73.0%)	59 (43.7%)
Anxiety		
High level (score ≥ 10)	171 (9.5%)	39 (28.9%)
Low level (score < 10)	1634 (90.5%)	96 (71.1%)
Occupational stress		
High level (score ≥ P_75_, 48)	479 (26.5%)	55 (40.7%)
Low level (score < P_75_, 48)	1326 (73.5%)	80 (59.3%)
Entrapment		
High level (score ≥ P_75_, 15)	472 (26.1%)	87 (64.4%)
Low level (score < P_75_, 15)	1333 (73.9%)	48 (35.6%)
Defeat		
High level (score ≥ P_75_, 24)	466 (25.8%)	85 (63.0%)
Low level (score < P_75_, 24)	1339 (74.2%)	50 (37.0%)
Interpersonal Needs		
High level (score ≥ P_75_, 51)	646 (35.8%)	74 (54.8%)
Low level (score < P_75_, 51)	1159 (64.2%)	61 (45.2%)

P_75_: the 75th percentile of the total scale score.

**Table 3 ijerph-18-11363-t003:** The association between psychosocial variables and suicidal ideation.

Psychosocial Variables	ORu (95% CI)	AOR (95% CI)	ORm (95% CI)
Depression			
High level	3.946 (2.759–5.644) **	3.691 (2.564–5.313) **	
Low level	Reference	Reference	
Anxiety			
High level	4.733 (3.134–7.150) **	4.600 (3.005–7.042) **	1.783 (1.089–2.920) *
Low level	Reference	Reference	Reference
Occupational Stress			
High level	2.020 (1.409–2.897) **	1.773 (1.222–2.571) **	
Low level	Reference	Reference	
Entrapment			
High level	6.050 (4.177–8.761) **	5.396 (3.696–7.876) **	2.064 (1.257–3.388) **
Low level	Reference	Reference	Reference
Defeat			
High level	5.751 (3.983–8.306) **	5.510 (3.788–8.014) **	2.572 (1.612–4.103) **
Low level	Reference	Reference	Reference
Interpersonal Needs			
High level	2.329 (1.635–3.317) **	2.332 (1.622–3.352) **	
Low level	Reference	Reference	

ORu: univariate odds ratios; AOR: odds ratios adjusted for age group, education level, marital status, number of children and work time outside of hometown; ORm: odds ratios obtained from multivariate logistic regression using significant variables of the univariate analysis as input. * *p* < 0.05, ** *p* < 0.01.

**Table 4 ijerph-18-11363-t004:** The association between the number of syndemic conditions and suicidal ideation.

	Number (%)	Had Suicidal Ideation	
		Number (%)	AOR (95% CI)
Have a Syndemic			
No (have no more than one psychosocial problem)	1104 (61.2%)	30 (22.2%)	Reference
Yes (have two or more psychosocial problems)	701 (38.8%)	105 (77.8%)	5.762 (3.773–8.802) **
Number of Syndemic Conditions			
No (have no more than one psychosocial problem)	1104 (61.2%)	30 (22.2%)	Reference
Low level (have two to three psychosocial problems)	451 (25.0%)	39 (28.9%)	3.149 (1.912–5.163) **
High level (have four or more psychosocial problems)	250 (13.8%)	66 (48.9%)	11.721 (7.322–18.761) **

AOR: odds ratios adjusted for age group, education level, marital status, number of children and work time outside of hometown. ** *p* < 0.01.

## Data Availability

Data are available on request due to privacy and ethical restrictions. The data presented in this study are available on request from the corresponding author. The data are not publicly available due to the protection of participants’ privacy.

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
