# Peer review of "The Mental Health and Syndemic Effect on Suicidal Ideation among Migrant Workers in China: A Cross-Sectional Study"

_ijerph, 2021, doi:10.3390/ijerph182111363_

Round 1

Reviewer 1 Report

Abstract

line 22. The study aimed..... Study doesn't aim the researcher aims.

Line 30. Multivariate logistic regression remained three psychosocial variables.... hard to understand what authors meant.

Line 31 various mental health issues integrated together.....tended to interact to amplify the effect on suicidal ideation. This sentence should be rewritten because  interaction was not tested  Simply,  people with more psychological problems were more likely to have suicidal ideation. Conclusion line 34-35 the words like magnifying and amplify are not appropriate.

Introduction section

Very long. Please try cut it down to relevant background information. Lacks purpose statement and planned hypotheses to be tested.

Numerous grammatical and incomprehensible sentence constructions .

The introduction also lacks logical sequences and connection between paragraphs. Needs extensive overhaul and extensive editing.

Material and methods

Line 116-118 should be rewritten.  Like the abstract and Introduction section, methods section also need extensive editing.

Results section.

Has similar issues poorly written and need extensive editing.

Why stepwise regression was used is not clear. Experts warn against using stepwise regression.

Discussion section equally suffers from same shortcomings. For example, line  285  wrong premise and conclusion from prevalence data that migrants  workers had more stress. Researchers have not compared locals and migrant workers. Excluding 218 locals was not appropriate. Would have been more interesting had they included this group and compared  them with migrant workers. Then you can answer questions regarding the relationship between suicide ideation and migration. For example, line 349 interact with each other?

Conclusions. Should be edited.  line 370. simultaneously interact to increase..... how?

Reviewer 2 Report

Zhang  and collaborators aimed to investigate mental health factors on suicidal ideation among migrants in a cross-sectional that included 1805 migrant workers in China. The research topic is very important for improving mental health and prevent suicide in migrant workers, especially considering the low number of suicide studies in this high-risk group. I have the following recommendations to improve the manuscript: 

Page 2, line 59. Some terms might be confusing. Consider removing the word “planning” as it could be mistaken with suicide planning 

Page 2, line 64. Consider changing the term “normal people” to controls.

line 72. Many other mental disorders increase the suicide risk besides major depression and anxiety, please mention them. 

Page 2 line 96. Please change the term “sexually transmitted infection patients” to “patients with sexually transmitted infections”, as the former term can be considered stigmatizing

Page 4 Line 169. Does the Job Stress Questionnaire have a validated Chinese version?

It should be considered a limitation that the suicidal ideation was evaluated by a single item of the questionnaire 

Page 5. Line 203. What was the cutoff to consider high or low levels? This information is mentioned only in the results, it should be in the methods as well. 

Overall, some grammar mistakes make the manuscript difficult to read. Please, review carefully the manuscript. 

Round 2

Reviewer 1 Report

Dear authors,

The manuscript needs extensive editing. Comment attached.
